# Developing a Multi-Criteria Decision Making approach to compare types of classroom furniture considering mismatches for anthropometric measures of university students

Pooya Khoshabi, Erfan Nejati, Seyyede Fatemeh Ahmadi, Ali Chegini, Ahmad Makui, Rouzbeh Ghousi *

Department of Industrial Engineering, Iran University of Science and Technology, Tehran, Tehran, Iran

* ghousi@iust.ac.ir

**Data Availability Statement:** All relevant data are within the manuscript and its Supporting Information files.

## Abstract

The mismatch between students' anthropometric measures and school furniture dimensions have been investigated in many countries. In Iran, collegians spend at least a quarter of the day hours at university in the sitting position, so it is essential to evaluate furniture mismatch among university students. In Iranian universities, the use of chairs with an attached table is widespread, while the study of mismatches in these chairs among the collegian community is rare. This study was aimed to compare and rank different classroom furniture types based on the mismatch between collegians' anthropometric measures and the dimensions of classroom furniture among Industrial Engineering students by developing a Multi-Criteria Decision Making approach in an integrated Methodology. The sample consisted of 111 participants (71 males, 40 females). Ten anthropometric measures were gathered, together with eight furniture dimensions for four types of chairs. Mismatch analyses were carried out using mismatch equations, and the Simple Additive Weighting method was used as a base method to solve the decision-making problem. The results indicated that Underneath Desk Height and Seat to Desk Clearance showed the highest levels of the match, while Seat Width presents the highest levels of low mismatch. According to the results, Type 1 and Type 3 were the best current classroom furniture. The Sensitivity Analysis was performed in two ways: changing the weights of criteria in nine scenarios and comparing the results with five other MCDM methods. The proposed MCDM approach can be used widely in furniture procurement processes and educational environments.

## 1. Introduction

Products are design based on specific functions that benefit users. However, their successful function is dependent on its usability for people [1, 2]. The furniture's function is to facilitate

**Funding:** The authors received no specific funding for this work.

**Competing interests:** The authors have declared that no competing interests exist.

learning and provide a convenient, stressless environment [2]. Anthropometry and ergonomics have been used to develop new furniture forms by incorporating adjustability to accommodate a broader range of people and populations [3]. The anthropometry data are essential for applying ergonomic principles for the design and improvement of different products [4].

Classroom furniture is a critical factor for the adoption of proper postures. Consequently, it has implications for the students' learning productivity [5] and students' comfort in the study environment [1]. In educational ambiances, anthropometry has become a crucial discipline. It can be used to provide pertinent students' anthropometric specifications, which can be used to put out critical information for school furniture design [4]. In developing countries, most classroom furniture has been found to have caused distractions and injuries to users due to wooden material and lack of quality [3].

Anthropometric measurements are a critical factor that should be taken into account in school furniture design, especially for the Asian population to have their anthropometric measurements regarding students so as it can be easy for designers who are intending to make ergonomic furniture [6]. In the design of classroom furniture, specific anthropometric measurements, such as Popliteal Height (PH), Knee Height (KH), Buttock-Popliteal Length (BPL) and Elbow Height (EH) are needed for the determination of the furniture dimensions which are essential to achieve the correct sitting posture [3, 7–10]. Gender differences should also be considered when designing classroom furniture based on anthropometric measurements [3, 11].

When the correct anthropometric data and sample population are not considered, a mismatch between anthropometric measures and dimensions of classroom furniture may occur [4]. A "mismatch" is defined as incompatibility between the classroom furniture dimensions and the students' anthropometric measures [7]. The mismatch is evaluated by mismatch equations that define the minimum and maximum limit of the furniture dimensions by using the anthropometric measures [12]. The mismatch could ultimately result in the development of musculoskeletal disorders within the students like low back pain [13] and other problems related to the learning process [4] because of poor classroom sitting posture [2]. So that some studies reported a large number of grade-school children and teenagers involved regular concerns of back and neck pain [14].

Prior to the 21$^{st}$ century, only a few studies had been done on the design of school furniture [10] but during the last few decades, there has been an increased concern about studies of school furniture and their match or mismatch to student's anthropometric measures [15]. Mismatches between school furniture dimensions and students' anthropometric measures have been investigated in many countries such as in South Korea [12], Greece [8, 9], Indonesia [15], Portugal [14, 16], United States of America [7], Palestine [10], Bangladesh [17], India [18], Chile [5, 13, 19, 20], and Iran [21, 22]. Table 1 shows the research goal and used techniques of mismatch studies in different countries.

Table 1 represents that descriptive statistics, linear regression, and hypothesis tests are frequently used in mismatch investigations. All the studies focused on mismatch levels using statistical analysis, while no mathematical model has been used to compare and rank types of classroom furniture based on the furniture mismatches in an integrated model.

In Iran, Dianat evaluated the mismatch between classroom furniture dimensions and anthropometric measures of 978 Iranian high school students in Kerman province in the Southeast of Iran, using nine anthropometric measurements and five dimensions to recognize any potential mismatch between them. The results showed a considerable mismatch in seat height (60.9%), seat width (54.7%), and desk height (51.7%) [22]. In 2018, Bahrampour investigated optimum seat depth using comfort and discomfort assessments among 36 university students aged 18 to 30 in Tabriz province in the Northwest of Iran. The findings suggested that

**Table 1. Literature review summary about mismatch studies in different countries.**

| Authors (Year) | Research Goal | Used Techniques |
|---|---|---|
| Castellucci et al. (2010) [13] | Evaluating mismatch between classroom furniture and anthropometric measures in Chilean schools | Descriptive Statistics & Kappa Coefficient |
| Dianat et al. (2013) [22] | Evaluating mismatch between classroom furniture dimensions and anthropometric measures of Iranian high school students | Descriptive Statistics |
| Castellucci et al. (2014) [19] | Applying different equations to evaluate the level of mismatch between students and school furniture in Chile | Descriptive Statistics & Kappa Coefficient |
| Macedo et al. (2015) [14] | Evaluating Match between classroom dimensions and anthropometry of Portuguese students for re-equipment according to European educational furniture standard | Descriptive Statistics |
| Yanto et al. (2017) [15] | Evaluating the Indonesian National Standard for elementary school furniture based on children's anthropometric measures | Descriptive Statistics, Independent T-test, Chi-Square Test & ANOVA |
| Bahrampour et al. (2018) [23] | Determining optimum seat depth using comfort and discomfort assessments | Descriptive Statistics, Chi-Square Test & Friedman Test |
| Halder et al. (2018) [17] | Assessment of mismatch values for recommending ergonomic considerations to design truck drivers' seats in Bangladesh | Univariate Linear Regression & Descriptive Statistics |
| Lee et al. (2018) [12] | Evaluating mismatch between furniture height and anthropometric measures for South Korean primary schools | Descriptive Statistics & Integer Linear Programming |

the 5th percentile of the BPL as an anthropometric criterion (40.2 cm) is the appropriate seat depth for the studied population [23].

In Iran, school students spend approximately one-third of their working hours at School in the sitting position [23] just like university students because Collegians spend at least a quarter of the day hours at university in the sitting position too, so it is essential to evaluate furniture mismatch among university students, but as mentioned, most of the studies have been done on school desks and chairs, while studies about the different type of chairs in college class-rooms are very limited. In Iranian universities, there are various types of furniture in different departments. However, the use of chairs with an attached table is widespread. In contrast, the study of mismatches in the chair with an attached table among the collegian community, especially in Iran, is rare.

As shown in Table 1, mismatch investigations focused on furniture mismatch levels using statistical analysis, but it is not sufficient to comprehensively compare types of furniture. So far, no mathematical model has been proposed to select classroom furniture types considering the furniture mismatches in an integrated framework, a critical research gap in the anthro-pometry field. Therefore, the below question can declare as a research question to fill the above research gap:

How can an integrated methodology lead to aggregating mismatch levels to select class-room furniture types regarding anthropometric considerations, furniture important criteria, statistical requirements, and mathematical models?

Thus, we want to use an appropriate approach in an integrated methodology to compare and select the best-used classroom furniture among Industrial Engineering students, so we used Decision Making science to compare and rank classroom furniture types. The practice of Decision Making is as old as the man [24], and the perception of a decision process involving a tradeoff amongst several criteria was put forward since centuries ago [25].

Multi-Criteria decision making (MCDM) is one of the most well-known branches of deci-sion making [26]. MCDM is divided into multi-objective decision making (MODM) and multi-attribute decision making (MADM). However, the terms MADM and MCDM are used to mean the same class of models. MODM peruses decision problems in which the decision space is continuous, while MCDM/MADM focuses on problems with discrete decision spaces

and try to find the optimal alternative amongst the existing decision alternatives concerning several criteria [27].

An MCDM method considers the preference structure of a Decision Maker (DM) and involves a value judgment. The DM's preferences will be incorporated in the decision model to support the alternative selection, and by doing so, the multiple criteria will be analyzed simultaneously [25]. On the other hand, using an MCDM method, the objectives are combined based on the DM's preferences. These preferences consist of the DM's subjective evaluation of the criteria, and the subjectivity is an intrinsic part of the problem and cannot be avoided [25].

There are many developed MCDM methods in the literature, and each method has its characteristics [27]. The MCDM methods may be categorized regarding their form of compensation for aggregating the criteria, which may be considered a kind of rationality. Hereon, two rationalities may be considered leading to compensatory and noncompensatory methods [28]. In a compensatory MCDM method, one criterion's disadvantage may be compensated for by the advantage in another criterion [25]. However, in a noncompensatory method, the mentioned disadvantage can not be compensated because, in a noncompensatory preference relation, no trade-off occurs [24]. Therefore, it seems that a compensatory MCDM method may be more appropriate to use for comparing classroom furniture based on the mismatch between anthropometric measures and furniture dimensions.

Various compensatory MCDM methods have been extended so far such as Simple Additive Weighting (SAW) method [29], Analytical Hierarchy Process (AHP) method [30–32], Analytical Network Process (ANP) method [33], Technique for Order Preference by Similarity to Ideal Solution (TOPSIS) [34–36], Multi-Attribute Utility Theory (MAUT) [37], Measuring Attractiveness by a Categorial Based Evaluation Technique (MACBETH) [38], Weighted Aggregated Sum Product Assessment (WASPAS) method [39, 40], VIseKriterijumska Optimizacija I Kompromisno Resenje (VIKOR) method [41–44], Additive Ratio ASsessment (ARAS) method [45], and Measurement Alternatives and Ranking according to Compromise Solution (MARCOS) [46]. Each of the methods uses numeric techniques to help decision-makers choose among a discrete set of alternative decisions. This is achieved based on the impact of the alternatives on specific criteria and thereby on the overall utility of the DM(s) [27] and Utility Function of DM(s), which can vary from one DM to another, define appropriate MCDM method to use.

MCDM methods have a wide range of applications in business and industries. MCDM applications are abundant in different fields of engineering and technology such as supplier selection [46, 47], portfolio selection [48], analysis of investments in construction [49], selection of transport resources in logistics [50], and machine tool selection [51]. While MCDM applications in Human Factors Engineering and Ergonomics are minimal. As represented in Table 2, there are few studies about MCDM application in Ergonomics. Most studies focused on human error and the work environment, while the MCDM used methods were TOPSIS, AHP, ANP, and DEMATEL method. There is only one study in the anthropometry field that investigates automobile seat comfort using a subjective framework [52]. Therefore, the contributions of this paper are 1. Proposing an integrated methodology for MCDM application in Anthropometry for the first time in the literature, and 2. Suggesting a mathematical model in an MCDM framework for classroom furniture types' selection based on aggregating mismatch levels for the first time in the literature.

So this study aimed to compare and rank different classroom types of furniture based on collegians' anthropometric measures and the dimensions of classroom furniture among Industrial Engineering students in Industrial Engineering Department and Ethics Department of Iran University of Science and Technology (IUST) by developing an MCDM approach in an integrated Methodology.

**Table 2. Literature review summary MCDM applications in ergonomics.**

| Authors (Year) | Research Goal | Field of Ergonomics | MCDM Used Methods |
|---|---|---|---|
| Jung et al. (1991) [53] | Resolving the conflict between different approaches for designing manual material handling | Manual Materials Handling | Heuristic method |
| Fazlollahtabar et al. (2010) [52] | Creating a subjective framework for seat comfort to help the automobile manufacturer provide their seats from the best producer regarding the consumers' idea | Anthropometry | AHP Entropy method TOPSIS |
| Chiu et al. (2016) [54] | Developing a latent human error analysis process to explore the factors of latent human error in aviation maintenance tasks | Human Error | Fuzzy TOPSIS |
| Ahmadi et al. (2017) [55] | Prioritizing the ILO/IEA Ergonomic Checkpoints' measures | Work Environment | ANP DEMATEL |
| Hsieh et al. (2018) [56] | Identifying the crucial human error factors in emergency departments in Taiwan using MCDM methods | Human Error | AHP Fuzzy TOPSIS |
| Wang et al. (2018) [57] | Assessing human error probability in high-speed railway dispatching tasks | Human Error | Fuzzy ANP |
| Mohammadfam et al. (2019) [58] | Investigating interactions among vital variables affecting situation awareness | Human Error | Fuzzy Dematel |
| Rossi et al. (2019) [59] | Providing the decision-makers of hospitals or diagnostic centers to select the best ultrasound device capable of optimizing workers' well-being | Work Environment | AHP |

In this study, the assumptions of the integrated methodology are the following:

1. The problem parameters were assumed deterministic. 2. The criteria were considered independently. 3. The chairs with the attached table were only for educational purposes.

## 2. Materials and methods

### 2.1. Study design

The study design consists of five phases (as shown in Fig 1). In the study phase, articles about the mismatch between anthropometric measures and furniture dimensions were studied and used mismatch equations, anthropometric measures, and furniture dimensions were extracted. In the experimental phase, a measurement guideline including furniture types and dimensions, measurement conditions, essential and significant anthropometric measures, minimum sample size, and mismatch equations were prepared according to the case study. Then, students' anthropometric measures and types of classroom furniture dimensions' were measured considering the mentioned measurement guideline. In the case study phase, other important criteria such as furniture material, furniture price, and furniture grace were defined by experts and their weights estimated by experts. Then, participants (University Students) as DMs filled the questionnaire to rank furniture dimensions from the most important (first rank) to the least important (last rank) based on their subjective evaluation (more details in section 2.8). In the statistical phase, sample anthropometric data were analyzed, and the needed statistics were calculated. Also, the mismatch percentage for every furniture dimension was extracted according to mismatch equations. In the decision making phase, considering questionnaires' results and experts' comments, weights of all criteria were calculated with the help of a suitable mathematical method. Then, the final decision making matrix was formed, including Alternatives (Furniture Types), Criteria (Furniture Dimensions and other Important Furniture Criteria), Weights of Criteria, and entries of the matrix (Match Percentage and values for other important criteria). Appropriate MCDM method was selected based on the case study and utility function of DM(s) to solve the decision making problem. Then, the classroom types of furniture were compared and ranked to determine the best existing used furniture. Finally, sensitivity analysis of results was performed in two ways (consist of comparing the

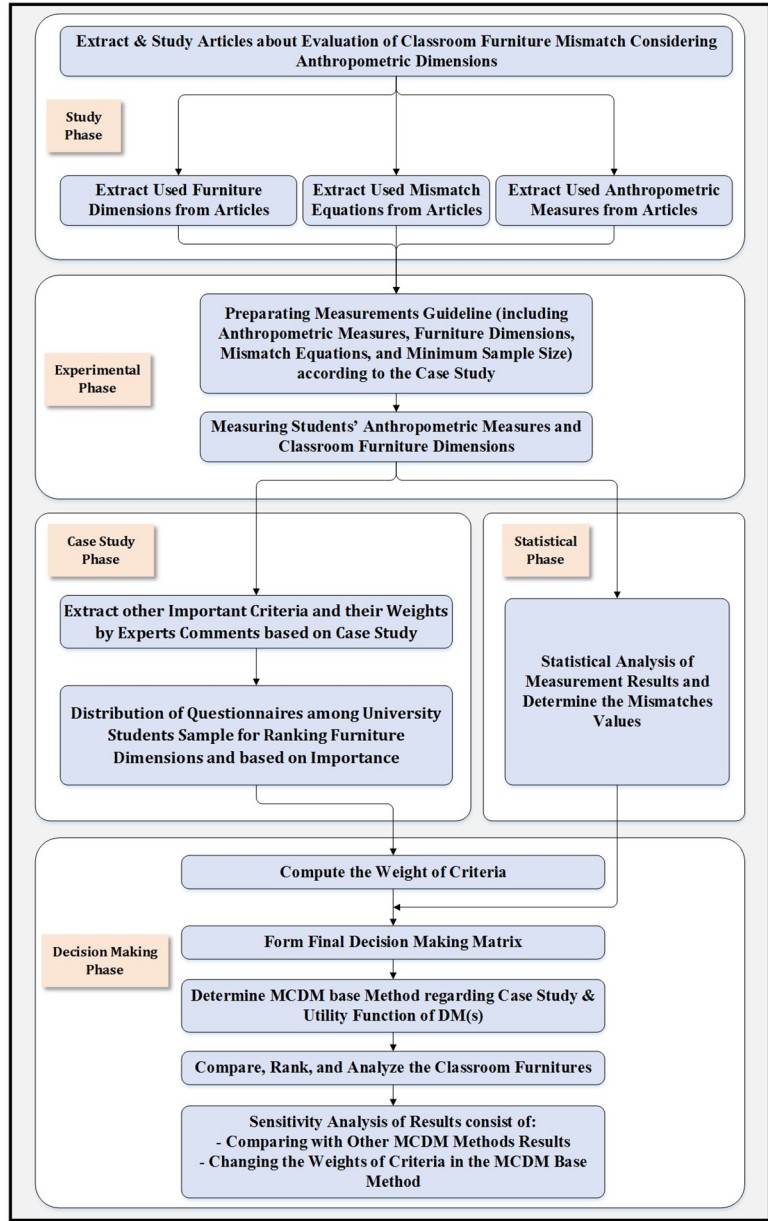

**Fig 1. Study design flowchart.**

results with other MCDM methods results and changing the weights of criteria in the MCDM base method).

## 2.2. Participants & sample size

The study population for this research was students from the Industrial Engineering Faculty of Iran University of Science and Technology (IUST) located in Tehran province. The experimental work has been approved by the Head of Industrial Engineering Faculty. The participants gave consent orally and take part voluntarily in the experiment work. The theoretical sample size is 106.8, considering a 50% prevalence of school furniture mismatch (p = 0.5) to obtain the largest sample [5], with 90% accuracy and 95% confidence interval. The student

population of the Industrial Engineering Faculty during 2020 January was 1119, where 329 (29.4%) of them were bachelor students, 741 (66.2%) were master students, and 49 (4.4%) were Ph.D. students with both male and female students. For each grade and gender, the students were selected using a stratified sampling method, subdividing based on grade and gender. Stratified sampling was used to select the participants since this method could guarantee that the samples represent specific sub-groups [15].

In this study, the final sample involved 111 students with ages ranging from 18 to 27 years old, with a mean age of 21.29 (SD = 1.88). Regarding the theoretical sample size, a sample size of 111 in this study would be sufficiently precise to estimate the degree of mismatch with a 95% confidence interval. The sample covered both gender and every degree from bachelor to Ph.D. degree. Table 3 presents the grade and gender distribution of students who participated in this study.

## 2.3. Measurements

Each participant was informed entirely about the aims of the research, the tools, the method, and voluntarily participated in the anthropometric data collection. The team conducting the experiment consisted of three men and one woman so that individuals of the same gender as the participants could perform the measurements. The measurements were carried out during students' free time between classes and were always carried out on the right-hand side of the participants for a greater scientific uniformity [60].

Stadiometer used to measure Stature and Anthropometer used to measure other anthropometric measures. Except for stature, all anthropometric measures were collected with university students in the sitting posture. They were instructed to sit in a non-adjustable chair with a flat wooden seat had a high backrest for reducing measuring error due to poor gesture of students [6], in such a way that their thighs were in full contact with the seat, their lower and upper legs were at right angles (knee bent at 90̊), their feet were placed on the floor, and their trunk was upright.

There was no excessive clothing such as socks, jackets, overalls, and shoes, which were allowed to be worn during measurements [6, 15]. To maintain the accuracy of stature measurement, braids, or hairstyles that might interfere with the readings were brushed aside or flattened as much as possible [3].

## 2.4. Anthropometric measures

The following anthropometric dimensions were measured and collected during this study, which are illustrated in Fig 2:

- Stature (S): The vertical distance from the floor to the top of the head (vertex) was measured while the subject stands erect, feet together and unshod, the head oriented in the Frankfort plane [6, 13, 22, 61].

**Table 3. Participants' distribution based on gender and degree.**

| Gender | Degree | | | |
|---|---|---|---|---|
| | **Bachelor** | **Master** | **Ph.D.** | **Subtotal** |
| Male | 22 | 46 | 3 | 71 |
| Female | 12 | 26 | 2 | 40 |
| Subtotal | 34 | 72 | 5 | 111 |

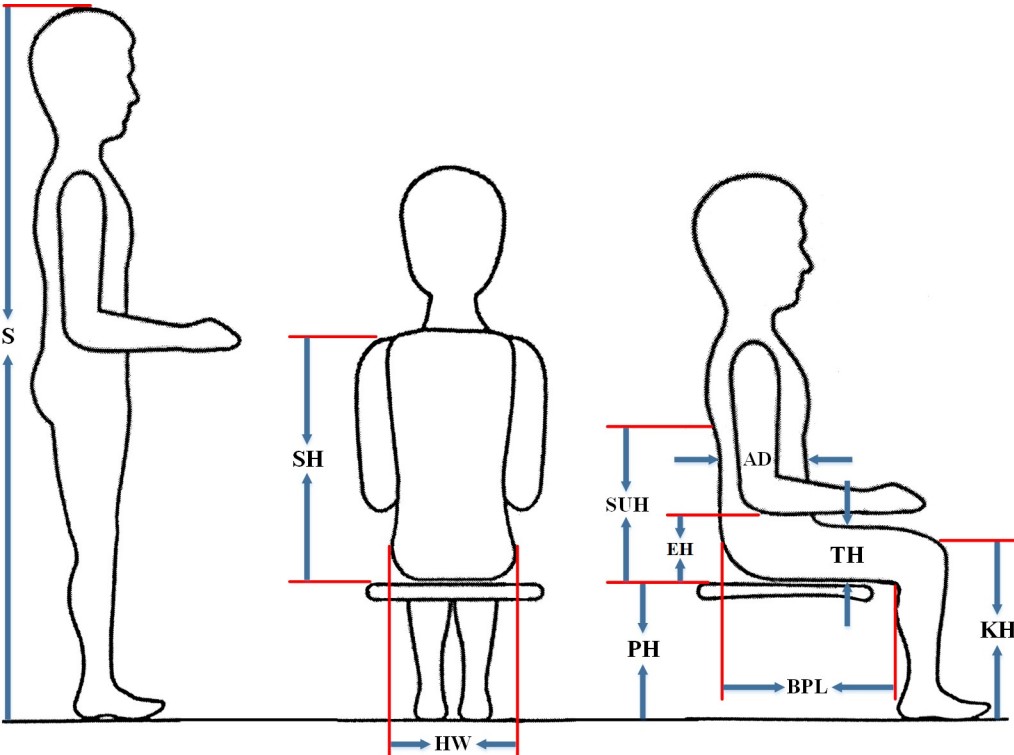

**Fig 2. Representation of the anthropometric measures.**

- Shoulder Height (SH): The vertical distance measured from the horizontal sitting surface to the top of the shoulder at the acromion with the arms hanging without restraint and the shoulders relaxed, in the sitting position with the back and legs angles 90° [3, 5, 12].

- Subscapular Height (SUH): The vertical distance measured from the lowest point (inferior angle) of the scapula to the subject's seated surface [5] while the hand was held straight from the shoulder forwards.

- Elbow Height (EH): The vertical distance from the bottom of the elbow tip to the subject's seated surface [62]. The elbow height was measured at an elbow flexion angle of 90° [3, 13].

- Abdominal Depth (AD): The maximum distance measured horizontally in a standard sitting position from the vertical reference plane to the front of the abdomen [17].

- Buttock-Popliteal Length (BPL): The horizontal distance from the posterior surface of the buttock to the popliteal surface, taken with a 90° angle knee flexion [13, 15].

- Popliteal Height (PH): The vertical distance measured from the foot-rest surface (supporting board) to the lower surface of the thigh immediately behind the knee, and the subject holds thigh and lower leg at right angles when seated [5, 10, 17].

- Knee Height (KH): The vertical distance from the foot resting surface to the top of the knee cap [62]. The KH is measured just above the patella at a knee flexion angle of 90° [3, 10, 12].

- Thigh Thickness (TT): The vertical distance measured from the horizontal sitting surface to the highest point on the thigh with an uncompressed soft tissue [3, 13, 22], when sitting with their thighs parallel and the feet in line with the thighs, at a knee flexion angle of 90° [3, 62].

- Hip Width (HW): The horizontal distance measured in the widest point of the hip in the sitting position [19, 22, 61].

  It should be noted that the Shoe Correction considered 3 centimeters (SC = 3 cm) for PH and KH in the mismatch equations.

## 2.5. Furniture dimensions

The following furniture dimensions, with the comprehensive descriptions, were gathered and are presented in Fig 3:

- Seat Width (SW): The horizontal distance from the outer right side of the sitting surface of the seat to the outer left side [8, 9].

- Seat Depth (SD): The distance from the back to the front of the sitting surface [7–9, 13].

- Seat Height (SH): The vertical distance from the floor to the highest point on the front of the seat [7–9].

- Desk Height (DH): The vertical distance from the floor to the highest point on the front of the seat [7–9, 19].

- Backrest Height (BH): The vertical distance from the top side of the seat surface to the highest point of the backrest [8, 9].

- Undersurface of Desk Height (UDH): The vertical distance from the floor to the lowest structure point below the desk [12, 19].

- Seat to Desk Clearance (SDC): The vertical distance from the top of the front edge of the seat to the lowest structure point below the desk (SH + SDC = Desk clearance) [13].

- Distance between the Desktop and the Backrest (DDB): The minimum distance measured horizontally from the back support to the steering wheel [17].

## 2.6. Mismatch equations

There is considerable variability about the mismatch equations in the literature, including one-way and two-way equations that can be used to test the mismatch between anthropometric measures and furniture dimensions. For the one-way equations only two categories or levels were defined: "Match" and "Mismatch" but for two-way equations, where both the minimum and maximum limits were considered, three categories were defined: (1) "Match" level, when the furniture dimensions are between the minimum and maximum limits; (2) "High mismatch" level, when the maximum limit of the equation is lower than the furniture dimension, indicating that the furniture dimension is higher than needed; and finally, (3) "Low mismatch" level, when the minimum limit of the equation is higher than the furniture dimension, indicating that the furniture dimension is lower than the recommended level [5]. In this study, eight furniture dimensions were evaluated, and for each furniture dimension, we used an appropriate mismatch equation as below:

1. Seat Width (SW): SW should be wide enough to be comfortable for someone who has the largest HW [63, 64]. [9] suggested that the SW should be at least 10% and at most 30% larger than HW. Considering this, Eq 1 shows the mismatch of SW [22].

$$1.1 \, HW < SW < 1.3 \, HW \tag{1}$$

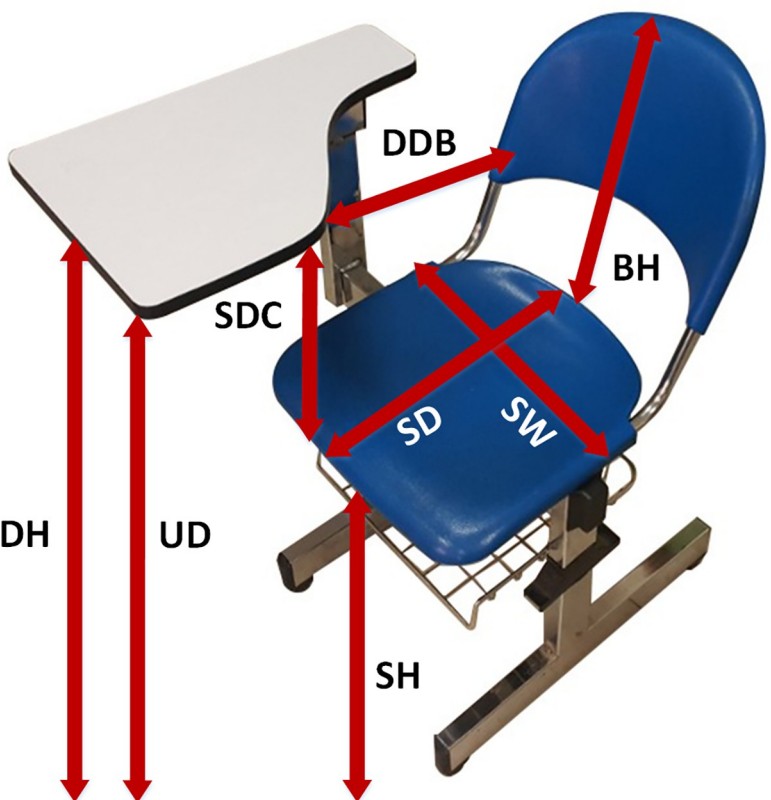

**Fig 3. Representation of the chair with an attached table dimensions.**

2. Seat Depth (SD): Regarding many studies, SD should be designed for the 5th percentile of BPL distribution to support the lumbar spine without compression [63]. Based on [7], lack of support of the lower thigh is caused when the SD is too shallow. We use Eq 2 as the mismatch equation for seat depth [10]:

$$0.80\text{BPL} < \text{SD} < 0.95\text{BPL} \tag{2}$$

3. Seat Height (SH): The mismatch equation with the regard of SH is calculated by Eq 3, which is related to PH. The equation below declares that SH should be lower than PH so that the lower leg forms 5°-30° angle relative to the vertical [7, 9].

$$\text{PHcos}30° < \text{SH} < \text{PHcos}5° \tag{3}$$

4. Desk Height (DH): Elbow height is a significant parameter for the desk height's mismatch equation [63, 65]. In studies, this equation consist of multiple parameters which is written as Eq 4 [10]:

$$\text{EH} + (\text{PH cos }30)° < \text{DH} < (0.8517\ \text{EH}) + (0.1483\ \text{SH}) + (\text{PH cos }5)° \tag{4}$$

5. Backrest Height (BH): BH is appropriate when it is below the scapula to move the trunk and arm correctly [64, 66] which is written in Eq 5:

$$SUH > BH \tag{5}$$

6. Undersurface Desk Height (UDH): UDH should be such that there is enough space between the UDH and the knee [10, 15, 63, 64]. In some researches, it is suggested that UDH should be at least 2cm larger than KH [7, 67]. Regarding this, the mismatch equation for UDH is shown in Eq 6 [10]:

$$KH + 2 < UDH \tag{6}$$

7. Seat to desk Clearance (SDC): SDC should be higher than TT to allow leg move freely [68]. In [7], it is suggested that at least a space of 2cm of TT should be considered as a lower limit for SDC. So, the mismatch equation is shown as Eq 7 [13]:

$$TT + 2 < SDC \tag{7}$$

8. Distance between the Desktop and the Backrest (DDB): In the literature, there is not any mismatch equation for DDB, but it is obvious that for a chair with attached table, DDB should be larger than AD, so in this way, the student can move freely, and the attached desk does not avoid him from moving while he is sitting on the chair. On the other hand, if DDB is greater than a specific limit, the user has to bend forward for writing, which leads to poor posture, so we suggested that the DDB should be at least 10% and at most 30% larger than AD for a chair with attached table, as Eq 8:

$$1.1\ AD < DDB < 1.3\ AD \tag{8}$$

## 2.7. Statistical analysis

The collected anthropometric data were analyzed with Minitab 19 and Microsoft Excel 2016. The Kolmogorov–Smirnov test (K-S test) was applied to check if the data set was well-modeled by normal distribution or not. The goodness of fit test was used to find the data distribution, which was not distributed normally, and T-test was used to compare males' and females' data. All data's statistical analyses were performed, considering a 99% confidence level.

The data were analyzed in terms of Mean, Standard Deviation (SD), Minimum (Min), Maximum (Max), 5th, 50th, and 95th percentile. All dimensions are in centimeter (cm). After the different equations were applied, categorical results (Low Mismatch, Match, and High Mismatch) were analyzed, and the mismatches for different types of furniture were compared using Radar and Bar Charts.

## 2.8. MCDM approach to compare classroom furniture

Modeling an MCDM problem needs to define alternatives, criteria, and the weight of each criterion. Alternatives ($A_i$; $i$ = 1,. . .,m) represent the different choices of action available to the DM(s). They are assumed to be screened, prioritized, and eventually ranked [27]. A criterion ($g_j$; $j$ = 1,. . .,n) is a tool constructed for evaluating and comparing alternatives according to the

point of view, which must be well-defined. Criteria divided into two categories: benefit criteria and cost criteria. Benefit criterion is a criterion that increases the utility by increasing its value, such as comfort, elegance, quality, and match percentage. Cost criterion is a criterion that decreases the utility by increasing its value, such as price, discomfort, and mismatch percentage.

This evaluation must take into account, for each alternative, all the pertinent attributes linked to the point of view considered. It is denoted by $a_{ij}$ that called the "performance" of $A_i$ according to the criterion $g_j$ [24]. Now, an MCDM problem can be expressed in a matrix format called "Decision Matrix" (D) that is an *(m×n)* matrix [27].

MCDM methods require that the criteria be assigned weights of importance. These weights ($w_j$>0) are normalized to add up to one [27]. The use of weights with the intensity of preference originates compensatory MCDM methods and gives the meaning of trade-offs to the weights of criteria. On the contrary, the use of weights with ordinal criterion scores originates noncompensatory aggregation procedures and gives the weights the meaning of importance coefficients [24]. The Sections 2.8.1 and 2.8.2 describe how the weights of criteria calculated for this study. Section 2.8.3 explains the final decision matrix with details, and the MCDM base method to solve the problem will be selected. Section 2.8.4 illustrates the sensitivity analysis procedure ultimately.

**2.8.1. Other important furniture criteria based on experts' comments.**   In the case study phase, other important furniture criteria such as furniture material, furniture price for buying, and furniture grace can be defined by experts based on the case study, and experts can estimate their weights. For this study, experts, including several faculty members, defined the furniture material (M) as the only other important furniture criterion for the chair with an attached table used in the university classrooms. The weight 0.08 considered for criterion M according to experts' comments.

**2.8.2. Ranking questionnaire and define weights of furniture dimensions.**   The participants (Industrial Engineering Department Students) as DMs filled the questionnaire to rank furniture dimensions from the most important (first rank) to the least important (last rank) based on their subjective evaluation. Experts defined the weight 0.24, 0.19, 0.14, 0.09, 0.08, 0.07, 0.06, and 0.05 for the first rank to the last rank of furniture dimensions, respectively. So, we had 111 rank number for every furniture dimension. Then, the rank numbers replaced by the defined weights to calculate the weights of furniture dimensions (first rank replaced by 0.24, second rank replaced by 0.19,. . ., eighth rank replaced by 0.05). Finally, 111 assigned weight for every furniture dimension were averaged to calculate each furniture dimension weight. All the calculations performed in Microsoft Excel 2016.

**2.8.3. Establish decision matrix.**   Final decision matrix can be formed (as Fig 4) based on Alternatives including Furniture Types ($A_i$; $i$ = 1,. . .,m), Criteria including Furniture Dimensions ($g_j$; $j$ = 1,. . .,b) and other Important Furniture Criteria ($g_j$; $j$ = b+1,. . .,n = b+k), Weights

**Fig 4. Decision matrix general form for this problem.**

of Criteria ($W_j$; $j = 1,...,$b) and entries of the matrix including Match Percentage ($m_{ij}$; $i = 1,...,$ m and $j = 1,...,$b) and values for other important furniture criteria ($c_{ij}$; $i = 1,...,$m and $j = b +1,...,n = b+k$).

Appropriate MCDM method can be selected based on the case study, the utility function of DM(s), and experts' comments to solve the decision-making problem. MCDM compensatory methods such as SAW, ARAS, MARCOS, VIKOR, WASPAS, MACBETH, TOPSIS, and MAUT seems suitable for such a problem, because one furniture dimension's disadvantage may be compensated for by the advantage in another furniture dimension. Between MCDM compensatory methods, we want to use a simple MCDM method with sufficient robustness against data measurement errors. Therefore, We utilized the SAW method as a base method for this study that uses the Weighted Sum Model (WSM) for calculating alternatives' utility based on the additive utility assumption of DMs [27].

SAW uses the Linear Scale Transformation method to normalize $a_{ij}$, while $a_j^{max}$ is the maximum entry in column j when $g_j$ is a benefit criterion and $a_j^{min}$ is the minimum entry in column j when $g_j$ is a cost criterion (Eq 9).

$$U(A_i) = \sum_{j=1}^{n} W_j u(a_{ij}) = \sum_{j=1}^{n} W_j(r_{ij}); \quad \begin{cases} r_{ij} = \dfrac{a_{ij}}{a_j^{max}} . \text{ Benefit Criterion} \\[2mm] r_{ij} = \dfrac{a_j^{min}}{a_{ij}} . \text{ Cost Criterion} \end{cases} \quad (9)$$

In this study, $U(A_i)$ can be written as Eq 10. The first term shows the weighted value of furniture anthropometric match, and the second term shows the weighted value of other important furniture criteria. $U(A_i)$ is obtained by adding two terms and shows DMs utility about furniture considering anthropometric criteria and other important criteria.

$$U(A_i) = \sum_{j=1}^{b} W_j u(m_{ij}) + \sum_{j=b+1}^{n=b+k} W_j u(c_{ij}) \quad (10)$$

It should be noted that the maximum value of the first term is $\sum_{j=1}^{b} W_j$, and the maximum value of the second term is $\sum_{j=b+1}^{n=b+k} W_j$, so these values need to be normalized for achieving an overall score between [0.1] that can be named Furniture Anthropometric Match Score (FAM Score) and Other Important Furniture Criteria Score (OIFC Score), respectively (Eqs 11 and 12).

$$\text{Furniture Anthropometric Match Score} = \sum_{j=1}^{b} W_j u(m_{ij}) \bigg/ \sum_{j=1}^{b} W_j \quad (11)$$

$$\text{Other Important Furniture Criteria Score} = \sum_{j=b+1}^{n=b+k} W_j u(c_{ij}) \bigg/ \sum_{j=b+1}^{n=b+k} W_j \quad (12)$$

The FAM Score has only an anthropometric look to ranking classroom furniture, while the OIFC Score has only a non-anthropometric approach to the problem. The Utility value of classroom furniture has a holistic/systematic approach for ranking classroom furniture regarding all the important furniture criteria. The higher value for the utility of a furniture type shows the more desirability in the DMs' point of view. Finally, the classroom types of furniture can be compared, ranked, and analyzed to determine the best existing used furniture.

**2.8.4. Sensitivity analysis.** The results of the base method (SAW) needs validation. We performed a sensitivity analysis in two ways: first, changing the weights of criteria in the SAW method and second, comparing the results with five other MCDM compensatory methods (TOPSIS [34], VIKOR [42] with $v = 0.7$, WASPAS [39] with $\lambda = 0.9$, ARAS [45], and MARCOS [46]). It should be noted that changing the weights of criteria carried out considering Kirkwood (1997) and Kahraman (2002) suggestions [69, 70].

## 3. Results

### 3.1. Anthropometric measures data

Table 4 presents descriptive statistics of anthropometric measures of 111 university students, which consist of 34 bachelor students and 72 master students and 5 Ph.D. students.

Normality of anthropometric measures was examined divided to Males and Females group using K-S test (with 99% confidence level). Obtained results showed that it was failed to reject the normal distribution of Males' Stature (p-value $\geq$ 0.150), Shoulder Height (p-

**Table 4. Anthropometric results.**

| Anthropometric Measures (cm) | Gender | Statistics | | | | | | |
|---|---|---|---|---|---|---|---|---|
| | | **Mean** | **SD** | **Min** | **Max** | **P5** | **P50** | **P95** |
| Stature | Male | 175.90 | 6.74 | 155.50 | 193.50 | 164.81 | 175.90 | 186.99 |
| | Female | 161.75 | 5.52 | 150.00 | 173.60 | 152.67 | 161.75 | 170.83 |
| | All | 170.80 | 9.29 | 150.00 | 193.50 | 154.80 | 170.80 | 185.36 |
| Shoulder Height | Male | 62.43 | 3.37 | 51.50 | 70.10 | 56.89 | 62.43 | 67.97 |
| | Female | 54.77 | 2.38 | 51.50 | 60.00 | 51.98 | 54.72 | 58.74 |
| | All | 59.67 | 4.78 | 51.50 | 70.10 | 51.82 | 59.67 | 67.53 |
| Subscapular Height | Male | 46.45 | 3.26 | 39.50 | 57.00 | 41.90 | 46.45 | 51.81 |
| | Female | 45.05 | 2.69 | 39.50 | 51.50 | 40.63 | 45.05 | 49.47 |
| | All | 45.93 | 3.11 | 39.50 | 57.00 | 40.80 | 45.93 | 51.05 |
| Elbow Height | Male | 24.5 | 2.58 | 17.70 | 30.80 | 20.26 | 24.5 | 28.74 |
| | Female | 23.65 | 2.07 | 20.50 | 29.00 | 20.25 | 23.65 | 27.04 |
| | All | 24.25 | 2.50 | 17.70 | 30.80 | 20.14 | 24.25 | 28.35 |
| Thigh Thickness | Male | 16.25 | 1.90 | 12.40 | 23.50 | 13.13 | 16.25 | 19.37 |
| | Female | 13.97 | 1.33 | 11.00 | 17.00 | 11.87 | 13.91 | 16.29 |
| | All | 15.44 | 2.03 | 11.00 | 23.50 | 12.10 | 15.44 | 18.78 |
| Abdominal Depth | Male | 22.70 | 3.34 | 16.20 | 32.90 | 17.58 | 22.72 | 29.36 |
| | Female | 20.64 | 2.64 | 16.10 | 26.50 | 16.68 | 20.48 | 25.16 |
| | All | 22.15 | 3.64 | 16.10 | 39.50 | 17.01 | 21.89 | 28.16 |
| Buttock-Popliteal Length | Male | 48.56 | 2.86 | 44.00 | 55.80 | 43.85 | 48.56 | 53.27 |
| | Female | 49.26 | 2.85 | 43.50 | 56.50 | 44.57 | 49.26 | 53.95 |
| | All | 48.82 | 2.85 | 43.50 | 56.50 | 44.12 | 48.82 | 53.51 |
| Popliteal Height | Male | 46.35 | 2.35 | 39.50 | 52.10 | 42.48 | 46.35 | 50.21 |
| | Female | 45.97 | 1.43 | 43.00 | 49.00 | 43.62 | 45.97 | 48.31 |
| | All | 46.22 | 2.56 | 39.50 | 52.10 | 42.00 | 46.22 | 50.43 |
| Knee Height | Male | 57.60 | 2.80 | 51.00 | 66.50 | 52.99 | 57.60 | 62.21 |
| | Female | 55.31 | 2.10 | 50.50 | 61.00 | 51.85 | 55.31 | 58.77 |
| | All | 56.79 | 2.79 | 50.50 | 66.50 | 52.21 | 56.79 | 61.38 |
| Hip Width | Male | 39.45 | 3.22 | 32.70 | 46.50 | 34.17 | 39.45 | 44.74 |
| | Female | 39.43 | 3.09 | 34.40 | 46.00 | 34.34 | 39.43 | 44.51 |
| | All | 39.53 | 3.28 | 32.70 | 49.30 | 34.14 | 39.53 | 44.92 |

value ≥ 0.150), Subscapular Height (p-value ≥ 0.150), Elbow Height (p-value ≥ 0.150), Thigh Thickness (p-value = 0.106), Abdominal Depth (p-value = 0.043), Knee Height (p-value ≥ 0.150), Buttock-Popliteal Length (p-value = 0.017), Popliteal Height (p-value ≥ 0.150), and Hip Width (p-value ≥ 0.150). Also, it was failed to reject the normal distribution of Females' Stature (p-value ≥ 0.150), Subscapular Height (p-value ≥ 0.017), Elbow Height (p-value ≥ 0.150), Abdominal Depth (p-value = 0.037), Knee Height (p-value ≥ 0.150), Buttock-Popliteal Length (p-value ≥ 0.150), Popliteal Height (p-value ≥ 0.067), and Hip Width (p-value ≥ 0.098).

Females' Shoulder Height (p-value ≤ 0.010) and Thigh Thickness (p-value ≤ 0.010) were not distributed normally. Using the Anderson-Darling test, it was found that Shoulder Height (p-value ≤ 0.022) and Thigh Thickness (p-value ≤ 0.059) variables were distributed lognormally. Dataset was normalized by performing Natural Logarithm on Shoulder Height and Thigh Thickness for Females.

An Independent t-test (with 99% confidence level) was performed to examine the differences in anthropometrics measures between Males and Females. Obtained results showed that there is a significant difference between Males and Females for Stature (p-value ≤ 0.001), Shoulder Height (p-value ≤ 0.001), Thigh Thickness (p-value ≤ 0.001), Abdominal Depth (p-value ≤ 0.001), and Knee Height (p-value ≤ 0.001) but there was not enough evidence to reject equality of Males and Females for Subscapular Height (p-value = 0.019), Elbow Height (p-value = 0.041), Buttock-Popliteal Length (p-value = 0.224), Popliteal Height (p-value = 0.276), and Hip Width (p-value = 0.792).

In this case, with a statistically significant difference, the Stature, Shoulder Height, Thigh Thickness, Abdominal Depth, and Knee Height of the Male students was higher than Female students.

## 3.2. Classroom furniture dimensions

Fig 5 presents a picture of four furniture types analyzed in this study. Fig 5 shows the measurement results of eight furniture dimensions for four types of chairs with the attached table too. Type 1 and Type 2 are the chairs that are used in the Industrial Engineering department. Type 1 is widely used in all classes in the Industrial Engineering department, and Type 2 is used in the exam halls scattered. Type 3 and Type 4 are the chairs that are used in the Ethics department. Type 3 is used in the classes of Ethics faculty for a long time. Type 4 is used in the classes of Ethics faculty lately, but this type is not as many as the previous one. The seat and backrest of Type 1 are made of plastic, while other types are made of wood. It should be noted that university male and female students use these types of furniture in the same condition.

## 3.3. Mismatch level

Fig 6 presents the mismatch level of furniture dimensions with eight anthropometric measures for all furniture types, divided into male and female groups. According to Fig 6, UDH shows a 100% match in all furniture types except Type 3 for males, and SDC presents at least 98% match for four different types of furniture. Type 1 and Type 3 indicate 100% match for BH, and Type 3 and Type 4 illustrate at least 97% match for males and at least 84% match for females in SH.

DDB presents the highest levels of high mismatch among furniture dimensions in four different types, so that at least 97% high mismatch was found in Type 1 and Type 3. In addition, Type 3, with at least 90% high mismatch in SD, has the maximum high mismatch in this dimension. SW presents the highest levels of low mismatch among furniture dimensions in four different types, so that at least 83% low mismatch was found in Type 2 and Type 4. Type 4

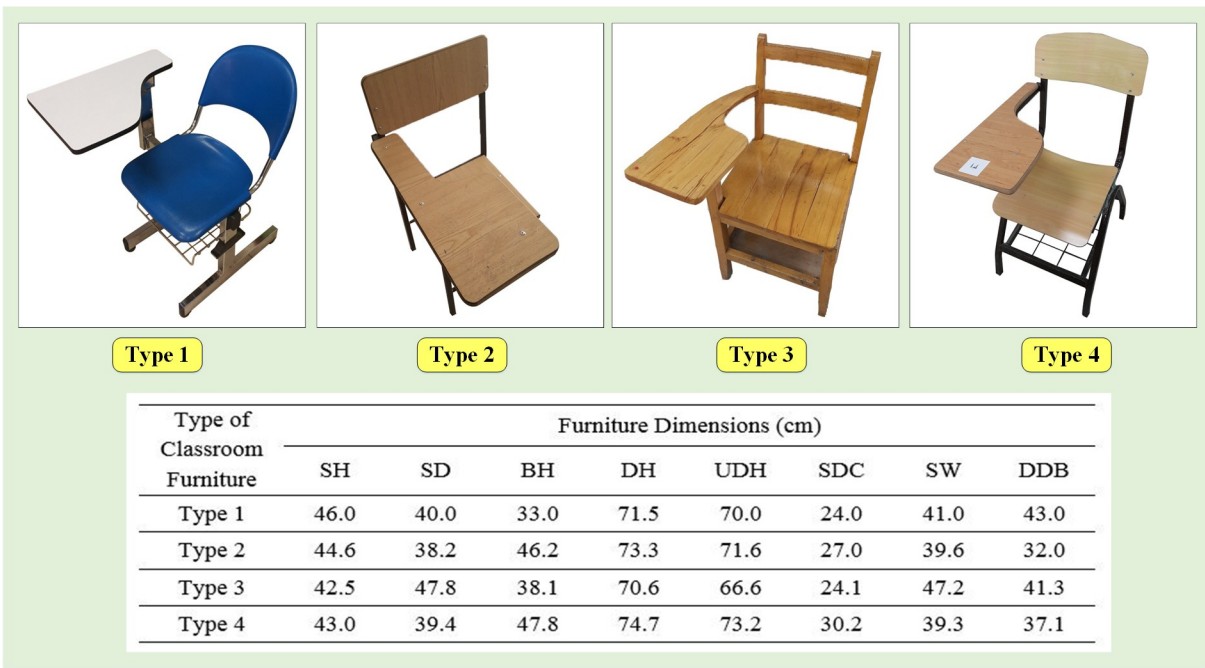

**Fig 5. Furniture dimensions results for four type of used chair with attached table.**

also indicates the highest value of low mismatch in DH. To improve the analysis, Fig 7 shows the Match Level for four types of furniture used in the decision matrix.

## 3.4. Solving the MCDM problem

**3.4.1. Final decision matrix.** Weights of criteria according to Section 2.8.2 for male, female, and all participants calculated. Experts determined desirability value for a wooden chair with an attached table (Type 2, Type 3, and Type 4) and plastic chair with an attached table (Type 1) about 0.350 and 0.700, respectively. Now, considering defined furniture dimensions (Section 2.5), other important furniture criteria (Section 2.8.1), the weights of criteria, desirability value for furniture material, and match values between anthropometric measures and furniture dimensions (Fig 7), final decision matrix according to Section 2.8.3 can be formed as Fig 8 that represents decision matrix for male, female and all participants before and after using Linear Scale Transformation method for normalization $a_{ij}$s (Eq 9).

**3.4.2. MCDM base method computations and results.** We used the SAW method to calculate alternatives' utility based on the additive utility assumption of DMs. According to Eq 10, the utility value of Type 1 considering male participants as DMs, can be written as below:

$$U(A_1) = (0.588 \times 0.105) + (1.000 \times 0.126) + (1.000 \times 0.178) + (1.000 \times 0.109) + (1.000 \times 0.095) +$$
$$(0.986 \times 0.103) + (0.478 \times 0.106) + (0.000 \times 0.098) + (1.000 \times 0.080) = (0.720) + (0.080) = 0.800$$

The 0.720 is the weighted value of furniture anthropometric match, and the 0.080 is the weighted value of other important furniture criteria that need to be normalized according to Section 2.8.3 (Eq 11 and Eq 12), by dividing to $\sum_{j=1}^{b} W_j = 0.920$ and $\sum_{j=b+1}^{n=b+k} W_j = 0.080$, respectively. So, in this case, FAM Score and OIFC Score were found 0.783 and 1, respectively.

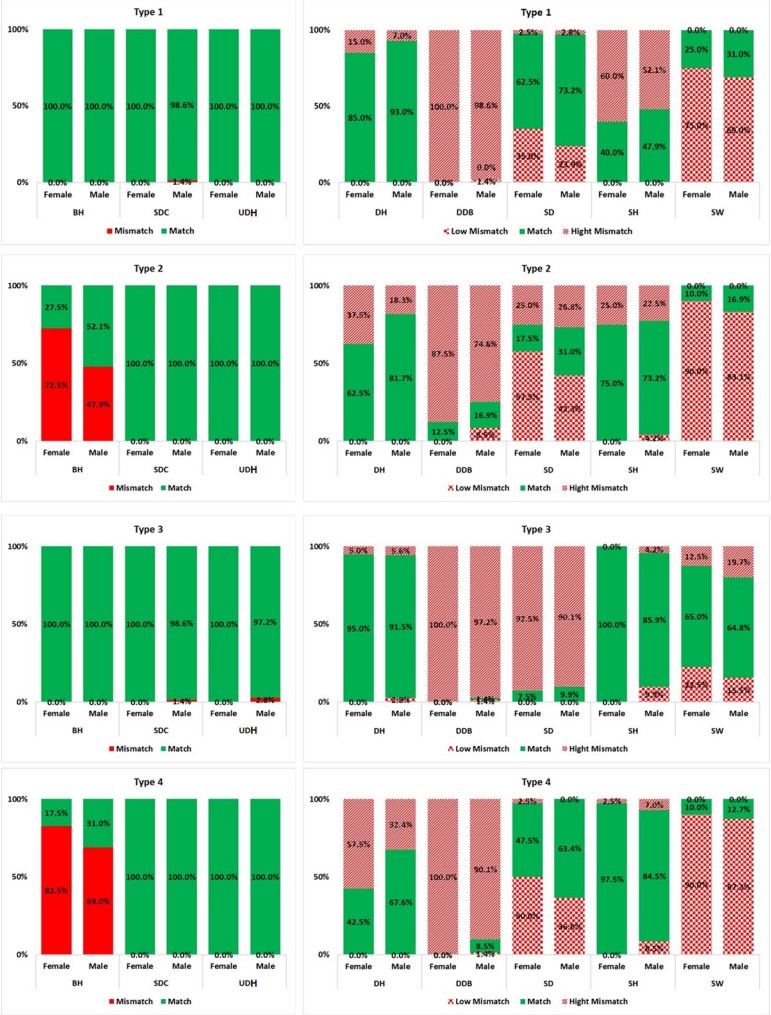

**Fig 6. Mismatch percentages considering the type of furniture and gender.**

According to Eq 10, the utility value of Type 3 considering all participants as DMs, can be written as below:

$$U(A_3) = (1.000 \times 0.117) + (0.130 \times 0.118) + (1.000 \times 0.175) + (1.000 \times 0.114) + (0.982 \times 0.092) +$$

$$(0.991 \times 0.100) + (1.000 \times 0.109) + (0.059 \times 0.095) + (0.500 \times 0.080) = (0.725) + (0.040) = 0.765$$

The 0.725 is the weighted value of furniture anthropometric match, and the 0.040 is the weighted value of other important furniture criteria that need to be normalized according to Eq 11 and Eq 12, by dividing to $\sum_{j=1}^{b} W_j = 0.920$ and $\sum_{j=b+1}^{n=b+k} W_j = 0.080$, respectively. So, in this case, FAM Score and OIFC Score were found 0.788 and 0.500. It means that the Type 3 Furniture Anthropometric Match Score is 0.788 that is the best score among current classroom furniture types. The Type 3 Other Important Furniture Criteria Score is 0.500, which is the second score.

Other utility values, FAM Scores, and OIFC Scores calculated as above and the results are shown in Fig 9.

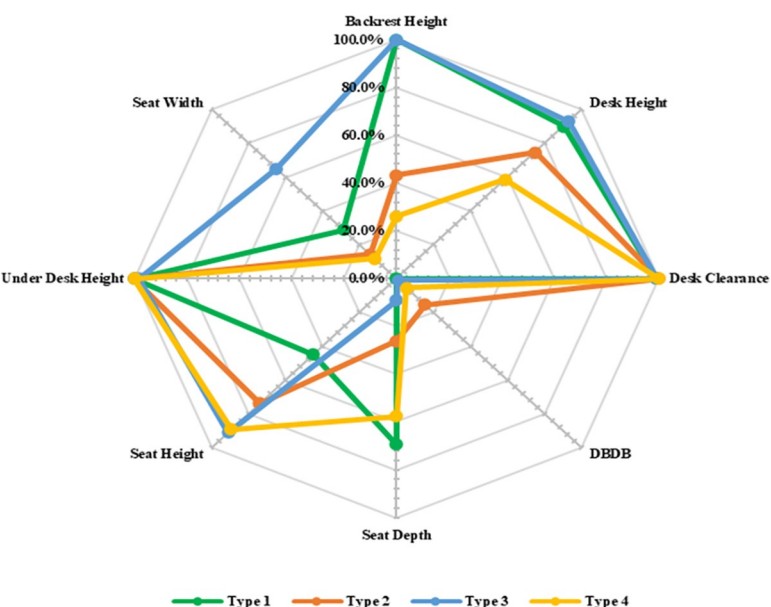

**Fig 7. Radar chart of match percentages considering the type of furniture.**

Fig 9 indicates that Type 1 furniture is the best existing chair with an attached table in this study, and Type 3 furniture is in the second rank. However, if only female participants considered as DMs, Type 3 furniture is the best existing furniture, and Type 1 will be second. In all conditions, Type 2 and Type 4 will be in the third and fourth rank, respectively. FAM Scores in Fig 9 represent that if OIFC Score ignored, Type 3 would be the best current furniture for

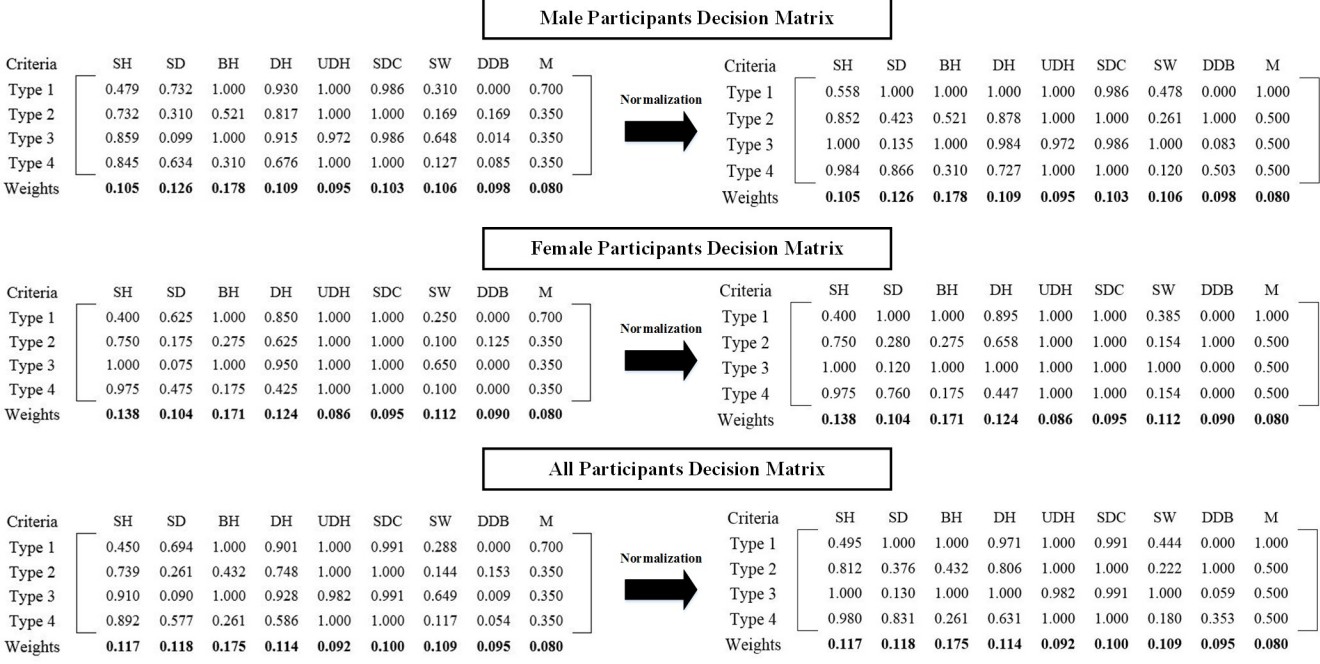

**Fig 8. Decision making matrices for male, female and all participants before and after normalization.**

| DMs | Furniture Utility Value | | | Furniture Anthropometric Match Score | | | Other Important Furntiture Criteria Score | | |
|------|------|--------|------|------|--------|------|------|--------|------|
| | Male | Female | All | Male | Female | All | Male | Female | All |
| Type 1 | 0.800 | 0.745 | 0.781 | 0.783 | 0.723 | 0.762 | 1 | 1 | 1 |
| Type 2 | 0.695 | 0.590 | 0.658 | 0.712 | 0.598 | 0.672 | 0.500 | 0.500 | 0.500 |
| Type 3 | 0.755 | 0.778 | 0.765 | 0.777 | 0.738 | 0.788 | 0.500 | 0.500 | 0.500 |
| Type 4 | 0.646 | 0.537 | 0.615 | 0.659 | 0.540 | 0.625 | 0.500 | 0.500 | 0.500 |

🟩 First Rank    🟨 Second Rank    🟧 Third Rank    🟥 Fourth Rank

**Fig 9. Results of MCDM method ranking, furniture utility value, and furniture scores.**

all participants in this study, and Type 1 will be the next. Generally, changing DMs or ignoring some of the important furniture criteria (including Anthropometric dimensions and other important furniture criteria) may lead DMs during the classroom furniture ranking process.

**3.4.3. Sensitivity analysis results.** *3.4.3.1. Changing the weights of criteria. B*ackrest Height (BH) with the largest weight of criteria ($W_3 = 0.175$) was considered the most influential criterion in this study. As shown in Table 5, the $\alpha_c$ values calculated for all criteria except BH ($\alpha_s = 1$). Then, the $\Delta x$ limits ($-0.175 \leq \Delta x \leq 0.8214$) and $\Delta x$ value ($\Delta x \cong 0.111$) were calculated. Then, the weights of criteria in nine scenarios calculated. Finally, the SAW method ran for each scenario to rank types of classroom furniture. This part of the sensitivity analysis only performed for all participants.

Fig 10 represents the results of the SAW method for nine scenarios. The results show that assigning different weights to criteria does not lead to the rank reversal. Utility values declare that Type 1 is the best current classroom furniture. Type 3 is in the second rank, while Type 2 and Type 4 are in the third and fourth ranks. Types of classroom furniture ranking do not change in all scenarios, but utility values change for all classroom furniture types due to increasing the weight of BH. Type 1 and Type 3 utility values increased, but Type 2 and Type 4 utility values decreased.

*3.4.3.2. Comparing results with other MCDM methods.* Table 6 represent the results of SAW method with five other MCDM methods for male, female, and all participants.

Obtained results confirmed that Type 1 is the best current classroom furniture type. The results have consistency within the MCDM methods except for WASPAS for all participants. The results for male participants have consistency within the MCDM methods except for VIKOR and WASPAS. For female participants, the results have consistency within the MCDM methods except for TOPSIS and WASPAS. Therefore, the SAW method had %100 consistency with ARAS and MARCOS methods.

**Table 5. Weights of criteria in nine scenario for sensitivity analysis.**

| $C_i$ \ $S_i$ | $S_0$ | $S_1$ | $S_2$ | $S_3$ | $S_4$ | $S_5$ | $S_6$ | $S_7$ | $S_8$ | $S_9$ | $\alpha_c$ |
|------|------|------|------|------|------|------|------|------|------|------|------|
| $C_1$ | 0.117 | 0.142 | 0.126 | 0.110 | 0.094 | 0.078 | 0.062 | 0.046 | 0.030 | 0.014 | 0.142 |
| $C_2$ | 0.118 | 0.143 | 0.127 | 0.111 | 0.095 | 0.079 | 0.063 | 0.047 | 0.031 | 0.015 | 0.143 |
| $C_3$ | 0.175 | 0.000 | 0.111 | 0.222 | 0.333 | 0.444 | 0.555 | 0.666 | 0.777 | 0.888 | 1.000 |
| $C_4$ | 0.114 | 0.138 | 0.123 | 0.108 | 0.093 | 0.078 | 0.063 | 0.048 | 0.033 | 0.018 | 0.138 |
| $C_5$ | 0.092 | 0.112 | 0.100 | 0.088 | 0.076 | 0.064 | 0.052 | 0.040 | 0.028 | 0.016 | 0.112 |
| $C_6$ | 0.100 | 0.121 | 0.108 | 0.095 | 0.082 | 0.069 | 0.056 | 0.043 | 0.030 | 0.017 | 0.121 |
| $C_7$ | 0.109 | 0.132 | 0.117 | 0.102 | 0.087 | 0.072 | 0.057 | 0.042 | 0.027 | 0.012 | 0.132 |
| $C_8$ | 0.095 | 0.115 | 0.102 | 0.089 | 0.076 | 0.063 | 0.050 | 0.037 | 0.024 | 0.011 | 0.115 |
| $C_9$ | 0.080 | 0.097 | 0.086 | 0.075 | 0.064 | 0.053 | 0.042 | 0.031 | 0.020 | 0.009 | 0.097 |

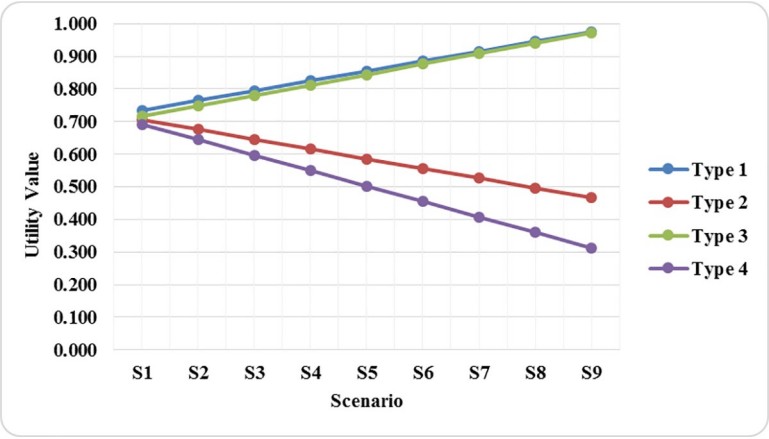

**Fig 10. Utility values chart regarding each scenario.**

## 4. Discussion and validation

Anthropometric results indicated that with a statistically significant difference, the Stature, Shoulder Height, Thigh Thickness, Abdominal Depth, and Knee Height of the Male students was higher than Female students. So as mentioned in previous studies [3, 11], gender differences should be considered in classroom furniture design based on anthropometric measurements. According to Table 2, the fifth percentile of the BPL in this study was 43.85 cm, 44.57 cm, and 44.12 cm for male, female and all participants, respectively, that is significantly greater than the fifth percentile of the BPL for 36 university students aged 18 to 30 in Tabriz province (40.2 cm) [23].

A mismatch equation for Distance between the Desktop and the Backrest (DDB) has been suggested to evaluate mismatch, especially for chairs with an attached table that are used in Iran Universities broadly. Although the equation needs to validate certainly in further researches, but it seems that DDB should be designed between two specific values that are a coefficient of Abdominal Depth (AD). In this case, the lower limit will guarantee to move freely, and the upper limit will prevent bending forward while writing on the table.

In this study, mismatch values for seat height were between 9% to 55% for different types of furniture, and mismatch values for desk height were between 7.2% to 41.4% for different types of furniture considering all participants, that are lower than the mismatches in seat height (60.9%) and desk height (51.7%) reported for 978 Iranian high school students in Kerman province [22].

In this study, the proposed MCDM approach used to compare different classroom furniture types based on furniture dimensions and other important furniture criteria. Using the SAW

**Table 6. Results of six MCDM compensatory methods.**

| | Male | | | | Female | | | | All | | | |
|---|---|---|---|---|---|---|---|---|---|---|---|---|
| | Type 1 | Type 2 | Type 3 | Type 4 | Type 1 | Type 2 | Type 3 | Type 4 | Type 1 | Type 2 | Type 3 | Type 4 |
| SAW | 🟩 | 🟧 | 🟨 | 🟥 | 🟨 | 🟧 | 🟩 | 🟥 | 🟩 | 🟧 | 🟨 | 🟥 |
| TOPSIS | 🟩 | 🟧 | 🟨 | 🟥 | 🟩 | 🟧 | 🟨 | 🟥 | 🟩 | 🟧 | 🟨 | 🟥 |
| VIKOR (0.7) | 🟩 | 🟨 | 🟧 | 🟥 | 🟩 | 🟧 | 🟩 | 🟥 | 🟩 | 🟧 | 🟨 | 🟥 |
| WASPAS (0.9) | 🟨 | 🟧 | 🟩 | 🟥 | 🟩 | 🟧 | 🟨 | 🟥 | 🟨 | 🟧 | 🟩 | 🟥 |
| ARAS | 🟩 | 🟧 | 🟨 | 🟥 | 🟨 | 🟧 | 🟩 | 🟥 | 🟩 | 🟧 | 🟨 | 🟥 |
| MARCOS | 🟩 | 🟧 | 🟨 | 🟥 | 🟨 | 🟧 | 🟩 | 🟥 | 🟩 | 🟧 | 🟨 | 🟥 |

method resulted in two furniture scores: Furniture Anthropometric Match Score (FAM Score) and Other Important Furniture Criteria Score (OIFC Score) that are weighted normalized values between [0.1]. FAM Score is an index that can indicate a furniture anthropometric match considering the weighted furniture dimensions' match. The index represents how match values belong to different furniture dimensions can aggregate based on users' opinions.

Fig 9 represented that classroom furniture ranking may vary by changing DMs because Type 1 is the best existing furniture for male and all participants while Type 3 is the best existing one for female participants, because as shown in Fig 8, male and female participants weighted furniture dimensions differently, and their match value for each furniture dimension is different. Therefore, the appropriate selection of users/DMs is so critical to achieving accurate ranking.

As discussed in [3], most classroom furniture has been found to have caused distractions and injuries to the users as a result of wooden material and lack of quality, in developing countries, so considering other important furniture criteria such as furniture material, furniture price, and furniture elegance can help DMs actively to compare and rank different types of classroom furniture for use, put aside and buy. On the other hand, ignoring other important furniture criteria may mislead the DMs in selecting the best classroom furniture. As shown in Fig 9, Type 1 that is a plastic chair, selected as the best existing furniture, while Type 3 that is a wooden one, will be the best existing furniture by ignoring furniture material criterion. Therefore, it seems that considering all the important furniture criteria can lead to the most accurate ranking for classroom furniture types.

In the sensitivity analysis, classroom furniture types ranking did not exchange by changing the weights of criteria, but utility values revolved (Fig 10). The correlation of SAW method results with ARAS and MARCOS method results was %100. The TOPSIS and VIKOR results had high consistency (not %100) with SAW method results. In contrast, the WASPAS method had different rankings for two alternatives (Type 1 and Type 3) for each group of participants that can be due to using Weighted Product Method (WPM) [39] in its utility function.

## 5. Conclusion

In this study, the mismatch between the anthropometric measures and classroom furniture dimensions were evaluated among 111 university students in Iran University of Science and Technology (IUST). The results indicated that UDH and SDC showed the highest levels of the match for four different types of furniture, while DDB presents the highest levels of high mismatch, and SW presents the highest levels of low mismatch among furniture dimensions. A mismatch equation has also been proposed for Distance between the Desktop and the Backrest (DDB) to evaluate mismatch in chairs with attached tables.

In this study, an MCDM approach has been proposed in an integrated methodology to compare and rank classroom types of furniture based on furniture dimensions' mismatches and other important furniture criteria values. The Proposed MCDM framework considering anthropometric guidelines, furniture important criteria, statistical requirements, and mathematical models is the novelty of this research, filling the research gap of MCDM applications in the Anthropometry field. This approach can be easily used widely in educational environments to evaluate classroom furniture types or in furniture procurement processes where anthropometric data are available. It seems that the usability and applicability of the proposed MCDM approach will be specified precisely in further researches. The limitations of the proposed methodology are 1. The problem parameters were considered deterministic. 2. Deterministic MCDM methods were used to solve the problem. 3. A simple method was used to weight the criteria.

Future studies can develop our approach in different fields of ergonomics, especially for various types of furniture with contributions such as considering parameter uncertainty, using different methods for weighting the criteria, and considering other important furniture criteria (Furniture Price, Furniture Elegance, etc.).

## Supporting information

**S1 Data.**
(XLSX)

**S2 Data.**
(XLSX)

## Author Contributions

**Conceptualization:** Ali Chegini, Ahmad Makui, Rouzbeh Ghousi.

**Data curation:** Pooya Khoshabi, Erfan Nejati, Seyyede Fatemeh Ahmadi.

**Formal analysis:** Pooya Khoshabi, Erfan Nejati, Seyyede Fatemeh Ahmadi, Ali Chegini.

**Investigation:** Pooya Khoshabi, Erfan Nejati, Seyyede Fatemeh Ahmadi.

**Methodology:** Pooya Khoshabi, Erfan Nejati, Seyyede Fatemeh Ahmadi, Ali Chegini.

**Project administration:** Pooya Khoshabi, Erfan Nejati, Seyyede Fatemeh Ahmadi, Ali Chegini.

**Software:** Pooya Khoshabi, Erfan Nejati.

**Validation:** Ahmad Makui, Rouzbeh Ghousi.

**Visualization:** Erfan Nejati, Ali Chegini.

**Writing – original draft:** Pooya Khoshabi, Erfan Nejati, Seyyede Fatemeh Ahmadi, Ali Chegini, Rouzbeh Ghousi.

**Writing – review & editing:** Pooya Khoshabi, Ali Chegini, Ahmad Makui, Rouzbeh Ghousi.

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
