## [Decision Letter · Decision Letter 0]

28 Jul 2020

PONE-D-20-21530

Developing a Multi-Criteria Decision Making Approach to compare types of Classroom Furniture Considering Mismatches for Anthropometric Measures of University Students

PLOS ONE

Dear Dr. Ghousi,

Thank you for submitting your manuscript to PLOS ONE. After careful consideration, we feel that it has merit but does not fully meet PLOS ONE’s publication criteria as it currently stands. Therefore, we invite you to submit a revised version of the manuscript that addresses the points raised during the review process.

We look forward to receiving your revised manuscript.

Kind regards,

Dragan Pamucar

Academic Editor

PLOS ONE

Journal Requirements:

3. We note that Figures in your submission may contain copyrighted images. All PLOS content is published under the Creative Commons Attribution License (CC BY 4.0), which means that the manuscript, images, and Supporting Information files will be freely available online, and any third party is permitted to access, download, copy, distribute, and use these materials in any way, even commercially, with proper attribution. For more information, see our copyright guidelines: http://journals.plos.org/plosone/s/licenses-and-copyright.

a)    You may seek permission from the original copyright holder of the Figures to publish the content specifically under the CC BY 4.0 license.

4. We noticed you have some minor occurrence of overlapping text with previous publications, which needs to be addressed. In your revision ensure you cite all your sources (including your own works), and quote or rephrase any duplicated text outside the methods section. Further consideration is dependent on these concerns being addressed.

5. Please clarify in your Methods section the full name of the ethics committee which approved the study, and state that they approved the study. Please also provide the approval number. Please also clarify how the students and experts gave consent.

6. We note that you have indicated that data from this study are available upon request. PLOS only allows data to be available upon request if there are legal or ethical restrictions on sharing data publicly. For more information on unacceptable data access restrictions, please see http://journals.plos.org/plosone/s/data-availability#loc-unacceptable-data-access-restrictions.

7. PLOS requires an ORCID iD for the corresponding author in Editorial Manager on papers submitted after December 6th, 2016. Please ensure that you have an ORCID iD and that it is validated in Editorial Manager. To do this, go to ‘Update my Information’ (in the upper left-hand corner of the main menu), and click on the Fetch/Validate link next to the ORCID field. This will take you to the ORCID site and allow you to create a new iD or authenticate a pre-existing iD in Editorial Manager. Please see the following video for instructions on linking an ORCID iD to your Editorial Manager account: https://www.youtube.com/watch?v=_xcclfuvtxQ

Reviewers' comments:

Reviewer's Responses to Questions

**Comments to the Author**

1. Is the manuscript technically sound, and do the data support the conclusions?

Reviewer #1: Yes

Reviewer #2: Yes

2. Has the statistical analysis been performed appropriately and rigorously? 

Reviewer #1: Yes

Reviewer #2: Yes

3. Have the authors made all data underlying the findings in their manuscript fully available?

Reviewer #1: Yes

Reviewer #2: Yes

4. Is the manuscript presented in an intelligible fashion and written in standard English?

Reviewer #1: Yes

Reviewer #2: Yes

5. Review Comments to the Author

Reviewer #1: Thank you for inviting me as a reviewer for manuscript titled Developing a Multi-Criteria Decision Making Approach to compare types of Classroom Furniture Considering Mismatches for Anthropometric Measures of University Students. In this study authors proposed MCDM framework for comparison of types of classroom furniture. The authors implemented SAW method for classroom furniture evaluation. The model is well explained. Methodology is clear. The paper is clearly, concisely, accurately, and logically written. I must congratulate to the authors for very quality research and effort for presenting results. The paper would be more exiting if you implement below improvements.

Specific comments

- Introduction section – Need to highlight the novelty of study in the introduction. Introduction should be clearly stated research questions and targets first.

- Why is the topic important (or why do you study on it)? What has been studied? What are your contributions? Why is to propose this particular MCDM framework?

- After that the authors need to discuss their contributions compared to those in related papers. The research gap and motivation should be clarified in the introduction (literature review) section. Authors should begin with the problem, the gap, then propose the research question and just after that say what they want to do to address that. Where is the gap? And you should clearly why it is a gap? Once again, if you say that it is a gap, then try to build a case for the gap. This part will help you for improvements that are required in the next comment.

- As a part of the literature review you can’t neglect the papers that are implementing MCDM related papers in your research. You are using hybrid SAW method. Why you have used SAW method and not for example MABAC, TOPSIS, MARCOS, COPRAS etc? To provide the stated answers, literature review will help you. You should present in the literature review with application of MCMD algorithms in different fields. There are numerous interesting papers that are presenting application of MCDM algorithm. I suggest authors to read and add below listed papers in the literature review: Si, A., Das, S., & Kar, S. (2019). An approach to rank picture fuzzy numbers for decision making problems. Decision Making: Applications in Management and Engineering, 2(2), 54-64.; Biswas, S., Bandyopadhyay, G., Guha, B., & Bhattacharjee, M. (2019). An ensemble approach for portfolio selection in a multi-criteria decision making framework. Decision Making: Applications in Management and Engineering, 2(2), 138-158.

- Validation of the results is missing. Validation should be presented based on variation of input parameters and comparison with existing MCMD methodologies.

- Add more deep discussion in section 3.4.2.

- Conclusion - Highlight the study novelty; Future directions; Show limitations of the proposed methodology.

Reviewer #2: Thank you for sending me for review the paper “Developing a Multi-Criteria Decision Making Approach to compare types of Classroom Furniture Considering Mismatches for Anthropometric Measures of University Students”. The topic is very interesting. The authors have presented the problem very detailed; I am giving support to the authors for investigation this topic. However, for the final acceptance of the work, it is necessary to make the following changes:

1. Insert the name of the applied method of multi-criteria decision making (SAW) in the keywords.

2. If abbreviations are used in the abstract, it is necessary to state their full name. In the main text of the paper, state the full title only for the first time and refer to the abbreviation. Use the abbreviation later.

3. In line 91-93 of the paper, the sources are listed where were investigated mismatches between school furniture dimensions and students' anthropometric measures. It is necessary to explain in 1-2 sentences what has been researched in the mentioned works and which methods have been applied.

4. After lines 218 and 249 (subtitles: 2.4. Anthropometric Measures and 2.5. Furniture Dimensions) it immediately starts with the enumeration. Before starting the enumeration, it is necessary to write one or two sentences of the introduction and point out what will be stated in the following text.

5. Line 280, 283, 287, 290, 293, 295, 299 and 302, subheadings 2.6.1 to 2.6.8 are used for enumeration. Use labels 1), 2) to 8).

6. In section 2.8.2. describe in detail the method by which the weight coefficients of the criteria were obtained (show step by step).

7. In section 2.8.3. explain why the SAW method was chosen (show it step by step).

8. For expressions where the calculation process is displayed, do not numbering the expression (such as expression 13). Delete number 13.

9. The sensitivity of the model was not performed in this paper. It is necessary to perform a sensitivity analysis. I suggest that the sensitivity analysis be performed at least in one way, such as changing the weight coefficients of the criteria (by favoring one criterion).

10. Bearing in mind that one of the first methods related to multi-criteria decision-making was used for ranking alternatives, in its original form, it is necessary to compare the results with 4-5 and if possible more other methods of older or newer date (TOPSIS, ELEKTRE, VIKOR, COPRAS, MABAC, CODAS, MAIRCA, TODIM, MARCOS, WASPAS ...).

11. It is necessary to edit the references - part of the references is incomplete (for example: line 596, line 597, line 600, line 609, etc.).

12. Most of the references are outdated. A very small number of references are more recent date (2018-2020), only one reference is from 2019 and 5 from 2018. References need to be supplemented with more recent sources (papers published from 2018 to 2020). Literature review is important for showing that authors are familiar with relevant research literature for the proposed field. I suggest that for example the sources listed in lines 141-146 be extended to newer ones such as:

- TOPSIS (A. Dobrosavljević, S. Urošević, Analysis of business process management defining and structuring activities in micro, small and medium – sized enterprises, Operational Research in Engineering Sciences: Theory and Applications, 2019, Vol. 2, no. 3, pp. 40-54);

- WASPAS (J. Mihajlović, P. Rajković, G. Petrović, D. Ćirić, The Selection of the Logistics Distribution Center Location Based on MCDM Methodology in Southern and Eastern Region in Serbia, Operational Research in Engineering Sciences: Theory and Applications, 2019, Vol. 2, no. 2, pp. 72-85)

- etc.

6. PLOS authors have the option to publish the peer review history of their article (what does this mean?). If published, this will include your full peer review and any attached files.

Reviewer #1: No

Reviewer #2: No

---

## [Author Response · Author response to Decision Letter 0]

24 Aug 2020

Response to Reviewers

Journal Requirements:

Thanks for this comment. The Title Page and Manuscript revised according to PLOS ONE's style requirements.

Thanks for this valuable comment. We used Grammarly (Premium Version) for language usage, spelling, and grammar, which is an English language digital writing tool. 

3. We note that Figures in your submission may contain copyrighted images.

Thanks for this comment. All the figures are original images except Figure 2. Figure 2 was a copyrighted image that has been edited. We removed the figure 2 from the submission and replaced it with a new, original, and unique figure which does not include copyright.

4. We noticed you have some minor occurrence of overlapping text with previous publications, which needs to be addressed. In your revision ensure you cite all your sources (including your own works), and quote or rephrase any duplicated text outside the methods section. Further consideration is dependent on these concerns being addressed.

Thanks for this precious comment. We checked the manuscript about overlapping with previous publications. We rephrased text or cited its sources.

5. Please clarify in your Methods section the full name of the ethics committee which approved the study, and state that they approved the study. Please also provide the approval number. Please also clarify how the students and experts gave consent.

Thanks for this comment. We clarified it (Line 222-224). It should be noted that the Iran University of Science and Technology does not have any ethics committee because it is a technical engineering university. In Iran, only universities of medical sciences have an ethics committee.

6. We note that you have indicated that data from this study are available upon request. PLOS only allows data to be available upon request if there are legal or ethical restrictions on sharing data publicly.

Thanks for this comment. The data submitted in two excel files entitled “Anthropometric Measures Results” and “Ranking Questionnaire Results”.

7. PLOS requires an ORCID iD for the corresponding author in Editorial Manager on papers submitted after December 6th, 2016. Please ensure that you have an ORCID iD and that it is validated in Editorial Manager.

Thanks for this comment. The ORCID iD validated for the corresponding author in PLOS ONE’s Editorial Manager.

Reviewers' comments:

Reviewer #1:

Thank you for inviting me as a reviewer for manuscript titled Developing a Multi-Criteria Decision Making Approach to compare types of Classroom Furniture Considering Mismatches for Anthropometric Measures of University Students. In this study authors proposed MCDM framework for comparison of types of classroom furniture. The authors implemented SAW method for classroom furniture evaluation. The model is well explained. Methodology is clear. The paper is clearly, concisely, accurately, and logically written. I must congratulate to the authors for very quality research and effort for presenting results. The paper would be more exiting if you implement below improvements.

Thanks for your comments.

Specific comments

- Introduction section – Need to highlight the novelty of study in the introduction. Introduction should be clearly stated research questions and targets first.

Thanks for this comment. We stated novelty of the study, the research gap, and research question in Line 94-97, Line 118-129, and Line 171-180.

- Why is the topic important (or why do you study on it)? What has been studied? What are your contributions? Why is to propose this particular MCDM framework?

Thanks for this comment. We answered to these questions in Line 94-97, Line 118-129, and Line 171-180.

- After that the authors need to discuss their contributions compared to those in related papers. The research gap and motivation should be clarified in the introduction (literature review) section. Authors should begin with the problem, the gap, then propose the research question and just after that say what they want to do to address that. Where is the gap? And you should clearly why it is a gap? Once again, if you say that it is a gap, then try to build a case for the gap. This part will help you for improvements that are required in the next comment.

Thanks for this valuable comment. We answered to these questions in Line 94-97, Line 118-129, and Line 171-180.

- As a part of the literature review you can’t neglect the papers that are implementing MCDM related papers in your research. You are using hybrid SAW method. Why you have used SAW method and not for example MABAC, TOPSIS, MARCOS, COPRAS etc? To provide the stated answers, literature review will help you. You should present in the literature review with application of MCDM algorithms in different fields. There are numerous interesting papers that are presenting application of MCDM algorithm. I suggest authors to read and add below listed papers in the literature review:

Si, A., Das, S., & Kar, S. (2019). An approach to rank picture fuzzy numbers for decision making problems. Decision Making: Applications in Management and Engineering, 2(2), 54-64.;

Biswas, S., Bandyopadhyay, G., Guha, B., & Bhattacharjee, M. (2019). An ensemble approach for portfolio selection in a multi-criteria decision making framework. Decision Making: Applications in Management and Engineering, 2(2), 138-158.

Thanks for this valuable comment. In Table 1 and Table 2, Mismatch investigations and MCDM applications in ergonomics considered. Some of MCDM applications in the different fields stated in Line 168-171. We spoke in Line 412-420 about why we used the SAW method. Other MCDM methods were used to validate the SAW method. We considered Biswas et al. (2019) article in our research.

- Validation of the results is missing. Validation should be presented based on variation of input parameters and comparison with existing MCDM methodologies.

Thanks for this valuable comment. Sensitivity Analysis performed in two ways: changing the weights of criteria and comparing the results with five other MCDM methods (TOPSIS, VIKOR, WASPAS, ARAS, MARCOS).

- Add more deep discussion in section 3.4.2.

Thanks for this comment. More Explanations stated in section 3.4.2.

- Conclusion -Highlight the study novelty; Future directions;Show limitations of the proposed methodology.

Thanks for this comment. We represent the study novelty in Line 640-645, future directions and research limitations in Line 646-653.

Reviewer #2:

Thank you for sending me for review the paper “Developing a Multi-Criteria Decision Making Approach to compare types of Classroom Furniture Considering Mismatches for Anthropometric Measures of University Students”. The topic is very interesting. The authors have presented the problem very detailed; I am giving support to the authors for investigation this topic. However, for the final acceptance of the work, it is necessary to make the following changes:

Thanks for your comments.

1. Insert the name of the applied method of multi-criteria decision making (SAW) in the keywords.

Thanks for this comment. SAW has been added to keywords.

2. If abbreviations are used in the abstract, it is necessary to state their full name. In the main text of the paper, state the full title only for the first time and refer to the abbreviation. Use the abbreviation later.

Thanks for this comment. It has been considered in the manuscript.

3. In line 91-93 of the paper, the sources are listed where were investigated mismatches between school furniture dimensions and students' anthropometric measures. It is necessary to explain in 1-2 sentences what has been researched in the mentioned works and which methods have been applied.

Thanks for this valuable comment. Explanations stated in Table 1 and Line 94-97.

4. After lines 218 and 249 (subtitles: 2.4. Anthropometric Measures and 2.5. Furniture Dimensions) it immediately starts with the enumeration. Before starting the enumeration, it is necessary to write one or two sentences of the introduction and point out what will be stated in the following text.

Thanks for this comment. Descriptions added in Line 262-263, and Line 293-294.

5. Line 280, 283, 287, 290, 293, 295, 299 and 302, subheadings 2.6.1 to 2.6.8 are used for enumeration. Use labels 1), 2) to 8).

Thanks for this comment. It considered in the Manuscript in Line 324, 327, 331, 334, 337, 339, 343, and 346.

6. In section 2.8.2. describe in detail the method by which the weight coefficients of the criteria were obtained (show step by step).

Thanks for this comment. Descriptions stated in Line 400-404.

7. In section 2.8.3. explain why the SAW method was chosen (show it step by step).

Thanks for this comment. It explained Line 413-420.

8. For expressions where the calculation process is displayed, do not numbering the expression (such as expression 13). Delete number 13.

Thanks for this comment. Number 13 deleted in line 524.

9. The sensitivity of the model was not performed in this paper. It is necessary to perform a sensitivity analysis. I suggest that the sensitivity analysis be performed at least in one way, such as changing the weight coefficients of the criteria (by favoring one criterion).

Thanks for this comment. Sensitivity Analysis performed in two ways. One way is changing the weights of criteria in nine scenario. 

10. Bearing in mind that one of the first methods related to multi-criteria decision-making was used for ranking alternatives, in its original form, it is necessary to compare the results with 4-5 and if possible more other methods of older or newer date (TOPSIS, ELEKTRE, VIKOR, COPRAS, MABAC, CODAS, MAIRCA, TODIM, MARCOS, WASPAS ...).

Thanks for this valuable comment. Sensitivity Analysis performed in two ways. One way is comparing the results with five other MCDM methods (TOPSIS, VIKOR, WASPAS, ARAS, MARCOS).

11. It is necessary to edit the references - part of the references is incomplete (for example: line 596, line 597, line 600, line 609, etc.).

Thanks for this comment. All the incomplete references edited.

12. Most of the references are outdated. A very small number of references are more recent date (2018-2020), only one reference is from 2019 and 5 from 2018. References need to be supplemented with more recent sources (papers published from 2018 to 2020). Literature review is important for showing that authors are familiar with relevant research literature for the proposed field. I suggest that for example the sources listed in lines 141-146 be extended to newer ones such as:

- TOPSIS (A. Dobrosavljević, S. Urošević, Analysis of business process management defining and structuring activities in micro, small and medium – sized enterprises, Operational Research in Engineering Sciences: Theory and Applications, 2019, Vol. 2, no. 3, pp. 40-54);

- WASPAS (J. Mihajlović, P. Rajković, G. Petrović, D. Ćirić, The Selection of the Logistics Distribution Center Location Based on MCDM Methodology in Southern and Eastern Region in Serbia, Operational Research in Engineering Sciences: Theory and Applications, 2019, Vol. 2, no. 2, pp. 72-85)

- etc.

Thanks for this valuable comment. Literature Review updated. There are 16 references for 2018-2020. Two references from 2020, six references from 2019, and 8 references from 2018. Also, the above new articles considered in our research.

While revising your submission, please upload your figure files to the Preflight Analysis and Conversion Engine (PACE) digital diagnostic tool. PACE helps ensure that figures meet PLOS requirements.

Thanks for this comment. All the figures checked with (PACE) digital diagnostic tool.

Editor Comments:

Thank you for submitting your manuscript to PLOS ONE. Your manuscript files have been checked in-house but before we can proceed we need you to address the following issues:

1) Thank you for confirming that you replaced Fig 2.

However, before we proceed can you please address the following queries about Fig 3 and Fig 5:

a) Where did you obtain Fig 3 and Fig 5 and have they been previously copyrighted?

b) If any of the images in Fig 3 and Fig 5 have been previously copyrighted, we require specific consent from the copyright holder to publish these images in PLOS ONE, under the CC BY 4.0 license.

Kind regards,

Agnes Magyar

PLOS ONE

All the figures are original images in the revised manuscript. Figure 3 is an original image that has been taken in the Industrial Engineering Faculty. Then we removed the background in Photoshop software. Then we represented furniture dimensions on it in Microsoft Visio 2016 software.

Figure 5 is an original image that its parts have been taken in the Industrial Engineering Faculty and the Ethics Faculty. Then we removed the backgrounds in Photoshop software. Then we represented images and furniture dimensions results’ in one image using Microsoft Visio 2016 software.

The images that have been taken in the Industrial Engineering Faculty & the Ethics Faculty are shown in "response to reviewers" file.

So, all the figures does not include copyright.

---

## [Decision Letter · Decision Letter 1]

3 Sep 2020

Developing a Multi-Criteria Decision Making approach to compare types of classroom furniture considering mismatches for anthropometric measures of university students

PONE-D-20-21530R1

Dear Dr. Ghousi,

We’re pleased to inform you that your manuscript has been judged scientifically suitable for publication and will be formally accepted for publication once it meets all outstanding technical requirements.

Kind regards,

Dragan Pamucar

Academic Editor

PLOS ONE

Additional Editor Comments (optional):

Reviewers' comments:

Reviewer's Responses to Questions

**Comments to the Author**

1. If the authors have adequately addressed your comments raised in a previous round of review and you feel that this manuscript is now acceptable for publication, you may indicate that here to bypass the “Comments to the Author” section, enter your conflict of interest statement in the “Confidential to Editor” section, and submit your "Accept" recommendation.

Reviewer #1: All comments have been addressed

Reviewer #2: All comments have been addressed

2. Is the manuscript technically sound, and do the data support the conclusions?

Reviewer #1: Yes

Reviewer #2: Yes

3. Has the statistical analysis been performed appropriately and rigorously? 

Reviewer #1: Yes

Reviewer #2: Yes

4. Have the authors made all data underlying the findings in their manuscript fully available?

Reviewer #1: Yes

Reviewer #2: Yes

5. Is the manuscript presented in an intelligible fashion and written in standard English?

Reviewer #1: Yes

Reviewer #2: Yes

6. Review Comments to the Author

Reviewer #1: All the reviewers' comments have been addressed carefully and sufficiently, the revisions are rational from my point of view, I think the current version of the paper can be accepted.

Reviewer #2: (No Response)

7. PLOS authors have the option to publish the peer review history of their article (what does this mean?). If published, this will include your full peer review and any attached files.

Reviewer #1: No

Reviewer #2: No

---

## [Editor Report · Acceptance letter]

7 Sep 2020

PONE-D-20-21530R1 

Developing a Multi-Criteria Decision Making approach to compare types of classroom furniture considering mismatches for anthropometric measures of university students 

Dear Dr. Ghousi:

I'm pleased to inform you that your manuscript has been deemed suitable for publication in PLOS ONE. Congratulations! Your manuscript is now with our production department. 

Kind regards, 

on behalf of

Dr. Dragan Pamucar 

Academic Editor

PLOS ONE